# We Are Also Metabolites: Towards Understanding the Composition of Sweat on Fingertips via Hyperspectral Imaging

Emanuela Marasco [1,*], Karl Ricanek [2] and Huy Le [1]

1. Center for Secure Information Systems, George Mason University, Fairfax, VA 22030, USA; hle28@gmu.edu
2. Computer Science Department, University of North Carolina Wilmington, Wilmington, NC 28403, USA; ricanekk@uncw.edu
* Correspondence: emarasco@gmu.edu

**Abstract:** AI-empowered sweat metabolite analysis is an emerging and open research area with great potential to add a third category to biometrics: chemical. Current biometrics use two types of information to identify humans: physical (e.g., face, eyes) and behavioral (i.e., gait, typing). Sweat offers a promising solution for enriching human identity with more discerning characteristics to overcome the limitations of current technologies (e.g., demographic differential and vulnerability to spoof attacks). The analysis of a biometric trait's chemical properties holds potential for providing a meticulous perspective on an individual. This not only changes the taxonomy for biometrics, but also lays a foundation for more accurate and secure next-generation biometric systems. This paper discusses existing evidence about the potential held by sweat components in representing the identity of a person. We also highlight emerging methodologies and applications pertaining to sweat analysis and guide the scientific community towards transformative future research directions to design AI-empowered systems of the next generation.

**Keywords:** sweat metabolites; biometrics; human identification





## 1. Introduction

Despite the increasing adoption of biometric technologies, the performance of existing systems is proven to be degraded by various covariates, and their security is dangerously compromised by spoofing [1–5]. Matching algorithms designed for comparing primary identifiers (i.e., face and fingerprint) yield lower genuine scores for racial and ethnic minorities, which not only results in a less accurate output but also makes these groups even more vulnerable [6]. For example, the accuracy of algorithms that compare face images is affected by skin tone. Thus, malicious individuals may concentrate on zero-effort attacks or impersonation attempts against this weak category. In particular, morphing attacks, by which the portraits in travel documents are forged, have been found to be more successful for Asian females [7]. Advanced 3D printers have also been successfully used to create inexpensive fake fingerprints and bypass authentication [8].

These limitations may be caused by how the biometric information is processed in the pictorial approach, in which authentication is based on extrinsic features encoded in the spatial domain. Exploiting only extrinsic or "skin deep" properties of a human can lead to discrimination against racial and ethnic minorities and women, which thereby also leads to inaccurate and less secure systems (e.g., facial recognition). On the other hand, intrinsic characteristics alone can be easily imitated (e.g., brain waves biometrics). To achieve higher reliability and accuracy, technology must also exploit intrinsic properties of a biometric trait by accessing its chemical content. Therefore, this research leverages the ability of selected compounds detectable in human sweat to discover more discriminative features for representing a person's identity [9,10].

Recently, interesting approaches capable of providing valuable information about individuals involving the chemical content of sweat (e.g., skin secretions) have been investigated. An individual has up to three million sweat glands. These glands are distributed over nearly the entire body of a person. Each gland contributes a unique mixture of chemical compounds [11]. Moreover, researchers have determined that the number of active sweat-glands count on a fingertip is highly reproducible [12]. According to Jelly, 2009, every time there is contact between persons, objects and locales, there is an exchange of physical information. This exchange of information from the finger is caused by amino acids associated with natural skin secretions. These secretions can be detected up to 40 years after contact, depending on the surface [13].

Several studies have discussed a variety of analytical methods with which to reliably detect metabolites in human sweat. These compounds seem to be related to a person's metabolic process and, thus, to their physical properties such as health status [14]. Furthermore, since at a given time, two individuals do not exhibit the same hormone levels, the concentrations of sweat components vary based on the person. Understanding the content of sweat on human skin may have various relevant applications (e.g., identification of people or infer their gender) and may benefit different fields (e.g., security, forensic, medical) [15–18]. The use of sweat in biometric technology is challenged by a variety of aspects, including how many metabolites are in a sample, how much sweat is required to detect them, how they vary, and whether the deposited sample and the capture process are repeatable.

Section 2 of this paper focuses on discussing realistic cases where sweat samples have been successfully extracted from fingerprints as well as forensics and medical applications of metabolites monitoring. Section 3 envisions benefits and challenges of using sweat as a biometric modality. Section 4 explores emerging directions to build AI-empowered systems based on sweat metabolites detection. Section 5 presents sweat-based recent technologies, and Section 6 draws conclusions.

## 2. Analysis of Sweat Samples from Fingerprints

Due to the established uniqueness and persistence of their visual pattern, fingerprints are an important and reliable source of evidence used to identify individuals in forensics and security applications such as criminal investigations and access control, respectively. Less attention has been given to the substances (e.g., sebum, sweat) left behind with the impression of a fingerprint [19]. Sweat is mainly composed of water, but also minuscule particles including minerals (e.g., sodium and magnesium) and metabolites (e.g., lactate and urea). Metabolites are produced based on metabolic processes generally controlled by hormones. Understanding what are the main factors impacting sweat composition is essential to its use as a potential biomarker in various contexts, such as human health.

To process sweat, one must first investigate two crucial aspects: (1) whether there is sweat on fingertips and samples can be collected, and (2) whether the acquisition of sweat samples from fingerprints across subjects is uniform. Recent studies have focused on analysing how amino acids are distributed in sweat collected from fingertips by using techniques that can separate the lipid content of human sweat from water. The bioaffinity sensing methodology, referred to as bioassay, was able to extract and measure the amount of amino acids in fingerprint content based on absorbance intensities.

Sweat as a research field is growing. An interesting study has explored sweat as a test for drug response [20]. Drug metabolites have been successfully identified in the sweat samples associated with latent fingerprints [21]. Findings also show that there is a significant correlation between drug response from sweat analysis and those based on salivary tests. Further research has explored the use of finger sweat to determine gender from fingerprints, a revolutionary approach that has recently exploited sweat metabolites [11]. Sweat has been used as an indicator of certain disease states [22,23]. The examination of chloride levels in sweat may provide an indication of cystic fibrosis; sweat can also aid the detection of inflammation [24]. Furthermore, analyzing sweat of

patients with renal failure has been linked to a significant rise in magnesium, calcium, and phosphate, indicating disease-specific changes in sweat ion concentrations [25].

## 2.1. Use of Sweat Metabolites in Forensics

Mass-spectroscopy-based methods, including bioaffinity systems, have allowed researchers to measure up to hundreds of metabolites in a single sweat sample, but due to the need for reagents, these approaches are inconvenient, not conducted in real-time, high in cost, and cause degradation of the sample. The technique separates metabolites using liquid chromatography (i.e., chemical reagents) and detects them based on their unique mass-to-charge ratio and induced fragmentation. Although matching latent fingerprints has been a universally accepted and reliable identification method, researchers have demonstrated that the pictorial comparison does not exploit the information content in a latent fingerprint to its full potential.

In 2015, Huynh et al. directed attention to the biochemical content in a fingerprint (i.e., concentrations of specific amino acids) using a biocatalytic assay rather than analyzing only the physical image [11]. Their work is the first proof of a system that can use the content of sweat left on a surface (i.e., latent) to estimate the gender of the originator. These researchers focused their analysis on the biochemical content in the fingerprint through a biocatalytic assay. In particular, multiple metabolites found in trace amounts of sweat, such as lactate, urea, and glutamate, were detected and the concentrations of each were considered. The method used a straightforward enzymatic cascade that exploits the colorimetric properties of the substrates that produce visible color changes. These biomarkers can be detected in sweat metabolites at wavelengths covered by imaging spectrometers. The content present in the sweat left behind, namely the amino acids, was also used to determine the gender of the originator based upon the quantity of specific metabolites which is theorized to be a byproduct of the hormone differences between males and females [11].

In 2018, Mindy et al. discussed the use of biocatalytic enzyme cascades to differentiate people based on lactate, urea, and glutamate metabolites detected in sweat [10]. Results confirm that the levels of all three markers sufficiently differ among people.

In 2017, Agudelo et al. proposed the use of sweat content as a mechanism for continuous authentication and tracking [26]. In this system, a sensor is placed at the points of skin contact with a device used to acquire sweat samples. The user's profile is built by continuously measuring sweat levels at various times of the day during a monitoring period. The biochemical input is converted into output signals that are then statistically analyzed to establish the identity of the person holding or wearing the device. This approach is contact based, and the process is slow, with no imaging utilized.

## 2.2. Use of Metabolites for Coronary Heart Disease Detection

Coronary heart disease (CHD) typically has a long asymptomatic pre-clinical period, and is not diagnosed until the individual experiences symptoms of a myocardial infarction or stroke, or worse, sudden death. Symptoms may be atypical, which often leads to a delay in diagnosis, particularly for women, who die from CHD at a rate of one in four nationally. There are strong indications that heart disease is more common among service members and veterans [27]. Stress, smoking, post-traumatic stress, and hypertension are known risk factors for heart disease and are more common among members of the military than the general population.

Existing preventive strategies based on classic risk factors do not capture the complex nature of CHD, with many CHD presentations occurring in the absence of traditional risk factors [22,27,28]. Furthermore, conventional CHD testing methods can take several days to weeks to produce the results. Interesting studies have compared the diagnostic utility of established nuclear and echocardiographic stress testing methodologies with newer techniques such as coronary computerized tomography and cardiac magnetic resonance imaging and highlight their inherent limitations in patients with underlying left ventricular

dysfunction. The reliability of single-photon emission computerized tomography (SPECT) is limited by the analysis regional wall motion abnormalities that can also be present in idiopathic dilated cardiomyopathy (DCM), and by cases in which patients are affected by with balanced ischemia [29].

Previous research has pointed out that metabolic disturbances may help with CHD diagnosis. Lower levels of selected metabolites are associated with increased risk of incident CHD. Recent studies have also successfully related circulating metabolites to CHD, including biomarkers that can be detected in eccrine sweat. These advances may promise a better understanding of CHD. Scientists examined whether metabolomics could be used to predict signs of CHD in people [22,23]. In one study, 141 circulating metabolites were examined for estimating the risk of CHD in a population of over 70,000 Europeans. Among these 141 metabolites under study, the researchers discovered 24 metabolites that were connected to CHD [22]. A subset of these 24 discriminative metabolites, including the amino acids methionine, glutamine and histidine, has been found in human eccrine sweat [30]. Twenty-four metabolites were significantly and independently associated with incident CHD. These metabolites were mainly acyl-alkyl-PCs and diacyl-PCs. In particular, 5 of these 24 metabolites were found to be inversely correlated with CHD risk. The strength of this correlation is comparable to those of classic risk factors. Although more research is required to investigate the connection between newly discovered metabolites and the development of CHD, these biomarkers can be detected at wavelengths covered by imaging spectrometers. Their impact has not yet been examined through image analysis.

### 3. Exploiting Sweat Metabolites as a Future Biometric Modality

Sweat is currently analyzed using methods that are not suitable for real-time applications and require specially trained operators. Existing bioassay systems can inspect the chemical content in the sweat of finger marks; however, this approach requires the samples to be treated with costly reagent kits that destroy metabolites. Active research is exploring a contactless manner with which to acquire the sweat metabolites from fingertips via hyperspectral imaging (HSI) [31,32]. HSI has gained a lot of interest in various fields, including agriculture and medical research [33–35]. HSI technology can capture rich spectral information for the purpose of object identification, chemical analysis, identifying materials, and even biometric application. A hyperspectral image is a three-dimensional hypercube over many contiguous spectral bands. Unlike traditional color images, the 3-D channels are feature rich due to the hundreds of wavelengths that are encoded across the pixel. The use of hundreds of wavelength responses across the region of interest may encode characteristics that are currently undetectable with traditional three-channel imaging systems.

Building a deeper profile of the identity makes the link between the genuine person and the digital representation stronger, and, subsequently, the system processing it more resilient to spoofing. Presentation attacks (PAs) that would challenge the security of existing fingerprint systems are easy to detect through HSI sweat analysis, combining visual and chemical characteristics. Attacks that use artificial sweat or skin-like materials lack sweat's biochemical components and, thus, will be easily identified as fake/artificial. Attacks that utilize sweat maliciously obtained from the target individual will also fail since the system operates by associating sweat's compounds with certain pixels.

Using an HSI perspective of sweat as biometric technology has the benefits of existing systems without the underlying drawbacks. In traditional fingerprint identification, unique visual (extrinsic) characteristics are derived from the ridges and compared with stored data. Since the technology is vulnerable to spoofing, liveness detection modules need to be integrated in the system. These algorithms exploit characteristics of vitality such as perspiration without linking them to an identity. Thus, attacks that use artificial sweat or sweat samples acquired from an impostor can succeed. Heart rate-based recognition relies on 101 features describing heart rate variability (HRV), an intrinsic property that is limited by performance degradation due to the influence of various physiological factors

(e.g., respiration) as well as by constrained acquisition, since the heartbeat sensor must be placed on top of a fingertip's vein [36]. Furthermore, the system can be easily spoofed by presenting to the sensor an artificial finger equipped with a pipe that pumps saltwater to simulate blood flow. Finger vein authentication is based on extrinsic properties of an intrinsic pattern, i.e., blood vessels underneath the skin of the fingers. Due to their sub-dermal nature, finger veins can only be captured using a near-infrared light beam shone on one side of the finger, highlighting the vein pattern. Although this technology is touchless and accurate, it is deceivable with a printout of the finger vein pattern [37]. Electrocardiogram (ECG) biometrics are intrinsic, accurate and resistant to stealing and spoofing; however, it is possible to exploit previously captured ECG signals and deceive the system [38,39]. Electroencephalography (EEG) biometrics record intrinsic electrical properties of a brain, but researchers have demonstrated the viability of attacks that imitate a user's mental reaction [27].

When biometric data refers to the chemical content of a trait that is processed in the hyperspectral domain, typical techniques for spoofing (e.g., artificial fingerprints made in silicon or gelatin, high-resolution photograph, printouts, etc.) would not work. Beyond liveness detection typically based on image processing of grey-scale or RGB biometric data, HSI analysis enables a variety of anti-spoofing approaches including blood flow pattern, since in hypercubes data more characteristics are visible compared to traditional images. Applying HSI analysis to biometrics is a new field and may aid the discovery of novel features such as statistics extracted by using the spectral component. To succeed, a spoof attack must reproduce exactly the pixelwise reflectance spectrum across a continuous range of wavelengths, a characteristic that is not captured by existing biometric systems.

Demographic differentials have been assessed in existing computer vision-based biometric technologies. In fingerprint systems, although recent studies have demonstrated the possibility of extracting gender clues from sensor-based fingerprint images, existing sensing technologies may limit what can be discovered from this biometric trait. The impact may vary based on the specific technology, for example, in minutiae-based matching systems, those biases are present. A recent study from A. Jain focuses on exploring demographic differentials in fingerprint recognition across four major demographic groups for two state-of-the-art (SOTA) fingerprint matchers operating in verification and identification modes [40]. Experiments on more than 15K individuals show that demographic differentials in SOTA fingerprint recognition systems decrease as the matcher accuracy increases.

Deep learning-based matchers may behave differently, thus, a related investigation about potential biases must be carried out. In such frameworks, even the typical relationship between image quality and matching accuracy may be questionable. Thus, the presence and the extent of biases require research efforts.

In recent chemistry literature, a technique to determine demographic biases (gender) based on a panel of sweat metabolites has been discussed. This technique separates metabolites using liquid chromatography (i.e., chemical reagents), and detects them based on their unique mass-to-charge ratio and induced fragmentation [11]. Mass-spectroscopy-based methods, including bioaffinity systems, have allowed researchers to measure metabolites in a single sweat sample, but due to the need for reagents, these approaches are inconvenient, not conducted in real-time, high in cost, and cause degradation of the sample. HSI is a novel methodology that involves the acquisition of chemicals and discovery biases may represent one of the main contributions of future research.

## 4. Analyzing Biochemical Content through Imaging: Where Are We?

Sweat pores have been previously detected through color changes of a specific polymer in contact with sweat, creating a map of the pores featured by a unique dotted pattern [41]. Recent research has pushed the boundaries of biometric sensing and imaging by capturing hypercubes of human fingers and analysed the related spectra through statistical machine learning [42]. This study considered whether alteration of the data may be derived from applying hand sanitizer right before the acquisition and the utility of the collected image

since such a product may deprive the skin from sebum and dissolve its lipid levels [42]. By focusing on the spectrum associated with hyperspectral images of human fingers, the authors investigated variations before and after the application of a commercial hand sanitizer. The images collected at Mason from 50 subjects and captured in a temporal range of 1, 10, and 25 min after participants used the product [43]. Subjects were campus students, their families, friends, or people who came to the lab interested to participate. Eligible participants were of an age greater than 18, without metabolic diseases, not pregnant, and not overweight. Subjects with known health issues, those under hormone therapy or those with cuts on their finger were excluded from the study.

The hyperspectral instrument used was the Resonon PIKA L camera that covers wavelengths from 378 nm to 1023 nm. It featured a spectral resolution of 2.1nm, for a total of 300 spectral channels with the image resolution of $400 \times 900$ pixels. Each collected hypercube is a multi-dimensional array of $400 \times 900 \times 300$. For each individual, the dataset contains three identical scans on the right-hand index fingers that make up to six hypercubes. Therefore, a total of 300 hypercubes were obtained from 100 individuals. They explored various signal processing techniques and found differences in the spectra generated from the data before and after the sanitizer product had been used. However, selected pre-processing approaches that account for and correct baseline shifting (i.e., SG and Detrend) can attenuate these unwanted changes. This analysis will be extended to incorporate potential changes due to the use of cosmetics (e.g., hand cream).

## 5. Recent Technologies Using Sweat Metabolites

Recently, there has been development in the technology used to detect sweat metabolites, nutrients, and biomarkers. In 2022, Komkova et al. developed the lactate biosensor of the lactate oxidase–Prussian Blue enzyme–nanozyme type. They utilized a siloxane-perfluorosulfonated ionomer composite membrane to immobilize the enzyme–nanozyme. This allows the biosensor to show flux independence in the whole range of physiological sweat secretion rates. Integrating with high-accuracy wearable electronic devices, the biosensors are capable of real-time remote monitoring of sweat lactate concentration and its secretion rate simultaneously [44]. In sweat, copper is a biomarker of Wilson's disease and liver cirrhosis. In the same year, Yang et al. developed a revolutionary wearable microfluidic nanosensor that can effectively identify and measure the amount of copper excreted in sweat. The sensor is created using advanced printing technology and boasts seamless integration with a wireless smartphone-based readout system. The device can be effortlessly applied to the skin and has the additional capability of actively inducing perspiration. This stimulation helps maintain a balanced concentration of heavy metals in relation to the sample volume and sweat rate. Their system solved common issues in sensors, such as sweat rate normalization and reliable continuous monitoring [45]. In another study, Laochai et al. detected cortisol in artificial sweat by developing the thread-based electrochemical immunosensor using immobilization of anti-cortisol on a L-cys/AuNPs/MXene modified conductive thread electrode. Due to the cortisol processes, the blocking of electron transfer, oxidation current towards the antigen–antibody binding interaction decreases, and cortisol is detected electrochemically. The immunosensor presented in this study provides exceptional sensitivity, reproducibility, and long-term storage stability. With the integration of the technology on a wristband, this device has the potential to be used as a wearable electrochemical sensor for sweat cortisol [46]. Wang et al. designed a wearable electrochemical biosensor that can analyze all essential amino acids and vitamins in sweat continuously, during physical exercise and at rest. Consisting of graphene electrodes and integrating with modules for iontophoresis-based sweat induction, microfluidic sweat sampling, signal processing and calibration, and wireless communication, the biosensor allows the assessment of the risk of metabolic syndrome and early identification of abnormal health conditions [47].

## 6. Conclusions

This paper discusses the new and challenging topic of detecting sweat metabolites through hyperspectral imaging to design the next-generation AI-empowered systems. HSI can generate a fine representation of an object by capturing its light reflectance pattern, which can be related to chemical composition. This technology holds potential to benefit the disciplines invested in spectroscopy (e.g., biochemistry). Applying HSI to analyze the content of sweat is non-invasive, fast and does not require the use of reagents, unlike the existing methods based on bioassays. The concentrations of the biochemical content in human sweat have been measured using imaging spectrometers that, although accurate, are expensive and time consuming.To date, no imaging has been applied yet. This paper directs the community to investigate how to exploit and link new findings in chemistry to AI, creating new representations based on biomarkers quantified in the hyperspectral domain. The proposed research will have lasting effects on the field, encouraging other researchers as well as industry to continue working towards a HSI sweat-based systems.

**Author Contributions:** E.M. has been investigating the potential of sweat as a biometric modality by exploring directions to derive features for identification, demographic and health information from it. She has researched previous work on sweat analysis where samples were obtained from fingertips. She has been experimentally inspecting the application of HSI to fingerprints to build next-generation AI-empowered systems (Sections 1–4 and 6). K.R. contributed to the main idea of applying HSI and provided valuable feedback about exploring chemicals through advanced imaging. He also guided discussions about biases and security challenges when using sweat as a biometric modality (Section 3). H.L. researched and contributed to summarize recent technologies based on sweat analysis (Section 5). He also provided feedback on sweat metabolites properties. All authors have read and agreed to the published version of the manuscript.

**Funding:** Marasco was partially funded by the NSF Award # 2036151.

**Data Availability Statement:** The HSI data mentioned in this paper was collected at Mason and is available to the research community.

**Acknowledgments:** The authors would like to thank Robin Couch of the Chemistry Department at Mason and his postdoc Allyson Dailey for valuable feedback on the biomarkers described in the paper. The authors appreciated the useful and encouraging discussion with Christopher DeFilippi, a specialist in Cardiovascular Disease at Inova Health System. The author thanks James L. Gulley and Charalampos S. Floudas at National Institutes of Health (NIH) for their enthusiasm about exploring metabolites for detecting COVID-19. Huy Le was an undergraduate student in biology at Mason while funded by the Office of Student Scholarship, Creative Activities, and Research (OSCAR) Program.

**Conflicts of Interest:** The authors declare no conflicts of interest.

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
