# Peer review of "We Are Also Metabolites: Towards Understanding the Composition of Sweat on Fingertips via Hyperspectral Imaging"

_digital, doi:10.3390/digital3020010_

Round 1
Reviewer 1 Report
The paper discussed the challenging topic of detecting sweat metabolites through hyperspectral imaging to design the next-generation AI-empowered systems by presenting the recent work of sweat analysis in medical and forensic applications, and sweat metabolites detection. More Interesting contents need to be provided for readers:
(1) Demonstrating the images of sweat metabolites from fingertips via hyperspectral imaging (HSI) for different persons, such as children, adult men, adult women and elders, to show the uniqueness of individual sweat metabolites.
2) Presenting the methods of machine learning related to implement the detection of sweat metabolite in Section 3
On the quality of the English language, converting long sentences into short sentences would be more beneficial for readers to understand the contents
Author Response
Thank you for your comments, please find attached the answers.
